# Visual Multitask Real-Time Model in an Automatic Driving Scene

**Xinwang Zheng** [1,2], **Chengyu Lu** [1,3], **Peibin Zhu** [2] **and Guangsong Yang** [2,*]

1   Chengyi College, Jimei University, Xiamen 361021, China
2   School of Ocean Information Engineering, Jimei University, Xiamen 361021, China
3   School of Machinery and Transportation, Southwest Forestry University, Kunming 650224, China
*   Correspondence: gsyang@jmu.edu.cn

**Abstract:** In recent years, automatic driving technology has developed rapidly, and environmental perception is one of the important aspects of the technology of automatic driving. To design a real-time automatic driving perception system with limited computational resources, we first proposed a network with faster reasoning speed and fewer parameters by using multitask learning and vision-based recognition technology, which can target the three tasks of traffic target detection, drivable road segmentation, and lane detection that need to be performed simultaneously. Based on the Apollo Scape dataset, the experiment results show that our network is superior to the baseline network in terms of accuracy and reasoning speed and can perform various challenging tasks.

**Keywords:** automatic driving; deep learning; multitask learning; image processing; lane line detection; target detection

## 1. Introduction

Due to the decreasing oil resources and people's demand for new technologies, new energy vehicles are rapidly popularized and bring more possibilities for automatic driving technology. In the automatic driving technology, the ability to perceive the environment is the most critical step to achieve the automatic driving technology that meets human expectations, which can help the vehicle to perceive the surrounding environment and control the driving direction or other controls of the vehicle independently. Object detection based on visual recognition is one of the crucial technologies to help vehicles avoid obstacles or pedestrians and comply with traffic regulations. It also can detect the lane line and the drivable area by the visual perception system and plan the corresponding driving path while guaranteeing for the driving safety.

Because there are various possibilities in the real-world environment, it requires that the auto drive system has high real-time and accuracy, and only in this way can the auto drive system bring more control safety than human beings. The same applies to the visual perception system of the auto drive system. Taking the current Advanced Driver Assistance System (ADAS) [1] as an example, the computing power of its applied hardware platform is very limited, which is caused by the problems of hardware cost and computing power miniaturization, which are also difficult to solve in the short term. Therefore, it is expected that there will be a network that can balance the requirements of real-time, high-precision, and multitask.

Compared to traditional image processing methods, neural network-based target detection and image segmentation models offer more possibilities. Recently, some noteworthy work on object detection has been put forward, such as RetinaNet [2]. To solve the problem of an unbalanced distribution of positive and negative samples in a single-stage, the loss function is improved in some references to make the performance comparable to that of the two-stage algorithm. CenterNet [3] applies an anchored detection box network in object detection in a single scene, Fast R-CNN [4] uses an area detection algorithm to provide an object box, and

YOLO [5–8] also uses a predefined prediction area method for target detection. It is a simple and efficient end-to-end network. Common segmentation networks are usually applied to the problem of driving region segmentation, such as UNet [9], which uses U-shaped network structure and has achieved good results in medical segmentation tasks. SegNet [10] and PSPNet [11] optimize the network structure of the segmentation model, and the segmentation accuracy has been further improved. In contrast, for lane detection or segmentation tasks, a stronger network is needed to better integrate the high-level and low-level features so as to consider the global structured context to enhance the segmentation details [12]. On the other hand, in the real-time auto drive system, it is usually impractical to run a separate network for each task [13]. Then, the multitask learning network provides a reliable solution to save computing costs. The network has been designed as a combination of a backbone network and multitasking head, with an overall presentation of one backbone and three branches.

The main contributions of our work can be summarized as follows:

a    We proposed a new scheme by using CSPNeXt as the backbone network for multitask learning networks, which achieves efficiency while simplifying the structural design.

b    We employed advanced network data enhancement techniques, such as mosaic filling and image blending during data pre-processing, which facilitated the generalizability of the model to different road scenarios.

c    We also optimized the loss function of the target detection task head by allowing it to match the detection frame classification and shared weight layers using a soft-label approach, which is used to improve the accuracy and speed of the target detection head network.

d    Experimental results showed that the proposed network outperforms the underlying network structure in terms of generalization.

## 2. Related Work

Some related works are listed in this section.

### 2.1. Traffic Target Detection in Real Time

In recent years, scholars in the field of deep learning have carried out a lot of research work in the direction of traffic target detection, which can be roughly divided into first-order networks and second-order networks [14]. Some works directly improve the network detection accuracy by optimizing the second-order network or adopting a larger network model, and some works try to train the single-stage lightweight network through the continuous optimization of the YOLO series network to achieve rapid real-time detection. The latter has become the research focus recently.

### 2.2. Drivable Areas and Lane Splits

The task of image segmentation is one of the hot research directions in image processing. FCN [15] is the first one that has made significant progress in semantic segmentation by fully using the depth learning method, which has led the research station in this field in a new direction. In order to improve the speed of lane line detection, HOU [16] proposed a lightweight network model using a self-attention distillation (SAD) module, which has less reasoning time but will lose some detailed information. The effect of lane line detection in complex environments such as lane line missing or no visual clues is not good. NEVEN proposed LaneNet [17], which regards the lane line detection process as an instance segmentation problem. The LaneNet model consists of two branches. The binary segmentation branch distinguishes the lane line and the background through pixel-by-pixel semantic segmentation. The pixel embedding branch decomposes the lane line pixels into different lane line instances and combines the results of the two branches to obtain the effect of instance segmentation, thus completing the lane line extraction. In addition, TABELINI proposed the PolyLaneNet model [18], which uses the deep learning network to regress the curve equation of the lane line and output a polynomial and confidence score for each

lane line. The real-time performance is high, but when the lane line is seriously blocked, the detection performance will be significantly reduced.

### 2.3. Multitask Learning

The goal of multitask learning is to improve the reasoning speed of the multitask model by sharing most of the network weight parameters compared to the single-task model. MultiNet [13] implements three major tasks in the deep learning network, namely scene classification, target detection, and semantic segmentation. YOLOP [19], based on the lightweight of the YOLO series network, applies multitask learning to the embedded autopilot equipment, realizing real-time performance on the edge computing platform. On this basis, HyBridNet [20] combined with BiFPN [21,22] to further improve the accuracy. The previous designs are based on convolutional networks. Since last year, self-supervised attention mechanism multitasking models for natural language processing using Transformer [23] networks have been applied to industrial scenarios, such as Vit [24] and Swin-T [25]. This method has been further applied in Tesla Motors [26]. Although this method has higher accuracy, its demand for computational power is also high.

### 3. Our proposed Methods

In this section, we describe the implementation of high-performance neural networks to jointly perform tasks such as detecting traffic targets, segmenting drivable areas, and detecting lanes. We focus on three works, i.e., designing the units of the backbone network structure, optimization of the different task heads, and designing the loss function.

### 3.1. Design Idea

Based on the encoder and decoder of YOLOP, we modified the model to improve the reasoning speed of the model. Inspired by RTMDET [27], we redesigned a powerful backbone network unit and designed a novel target detection head using category separation and box separation detection heads. In addition, influenced by YOLOPV2 [28], we conducted experiments on the separation of the backbone network feature layers for segmentation tasks for specific datasets, and we found that the three branches of the task head are uniform for feature extraction, especially for lane line detection and drivable area segmentation, and, if the feature layers of both are input separately, it will lead to more difficulty for the model to learn effective semantic features. The segmentation accuracy is improved by optimizing the training parameters and other methods.

### 3.2. Network Architecture

Figure 1 shows the structure of the proposed network. The system consists of a backbone network for extracting features from the input image and three task-solving docks for the matching task.

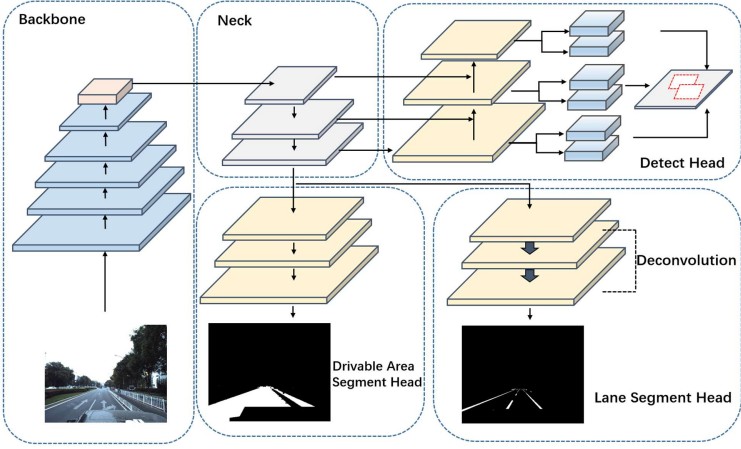

**Figure 1.** Schematic of network structure.

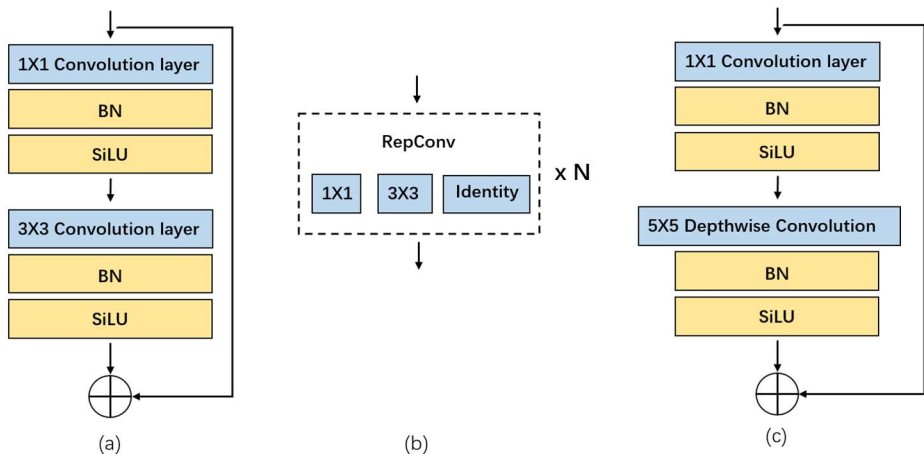

**Figure 2.** Structure diagram of Darknet Block.

As shown in Figure 2a, the BasicBlock consists of $1 \times 1$ and $3 \times 3$ convolution layers followed by the batch normalization and SiLU activation function.

Figure 2b is the structure of RepConv, which mainly uses re-parameterized blocks based on BasicBlock and is similar to YOLOv6, YOLOv7, and PPYOLO-E. However, the training cost of re-parameterization is high and difficult to quantify, so other methods are needed to compensate for quantization errors.

Figure 2c is RTMDet referenced the recently popular ConvNeXt and RepLKNet methods and added a large kernel deep convolution to BasicBlock, called the CSPNeXt block, which can improve the receptive field of a single convolution module in the block or learn enough features, as shown in Table 1 below.

**Table 1.** Convolution core size and performance test table.

| Kernel Size | mAP50:95 | Speed (FPS) |
| --- | --- | --- |
| YOLOP | 76.8 | 90 |
| CSPNeXtBlock $3 \times 3$ | 77.2 | 133 |
| CSPNeXtBlock $7 \times 7$ | 77.8 | 107 |
| CSPNeXtBlock $5 \times 5$ | 77.5 | 126 |

The speeds in all above tables are represented by frames per second (FPS). The main modules of the core network are composed of CSPLayer-T and SSP, as illustrated in Figure 3. The SSP layer is connected by the residual from the convolution layer and three max pooling layers and, finally, the output layer. SSP can enhance the receptive field of the dorsal network without reducing performance. CSPLayer-T is made up of the volume layer and CSPNeXtBlock, which were combined by channel attention.

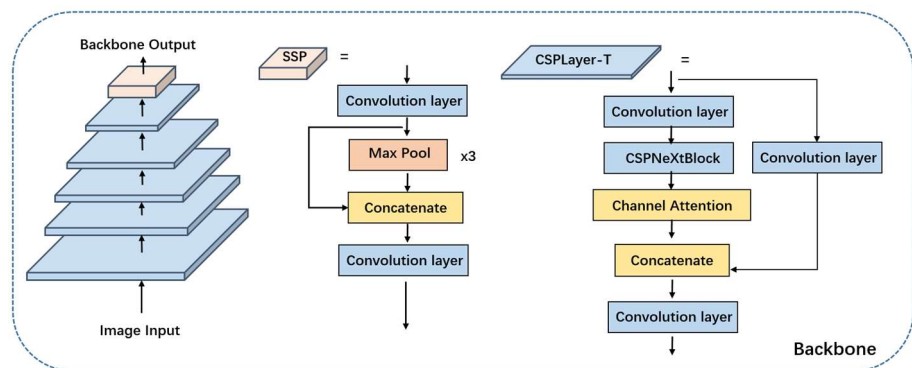

**Figure 3.** Backbone network architecture.

### 3.3. Task Headers

As mentioned above, we designed three different decoupling docks for each task. Similar to YOLOvX, we used a multiscale detection scheme decoupled by classification and regression branches. First, we used a bottom-up path aggregation network (PAN) to better extract semantic features at different scales. By combining PAN and FPN features to form a dual pyramid structure, global semantic information is fused with these local features to obtain richer, high-level semantic information, and then the multiscale feature maps output from the Neck structure are detected.

The Neck structure is shown in Figure 4 below. Its main component module is CSPLayer-N, because it is adopted before input and does not use the CSPNeXtBlock layer.

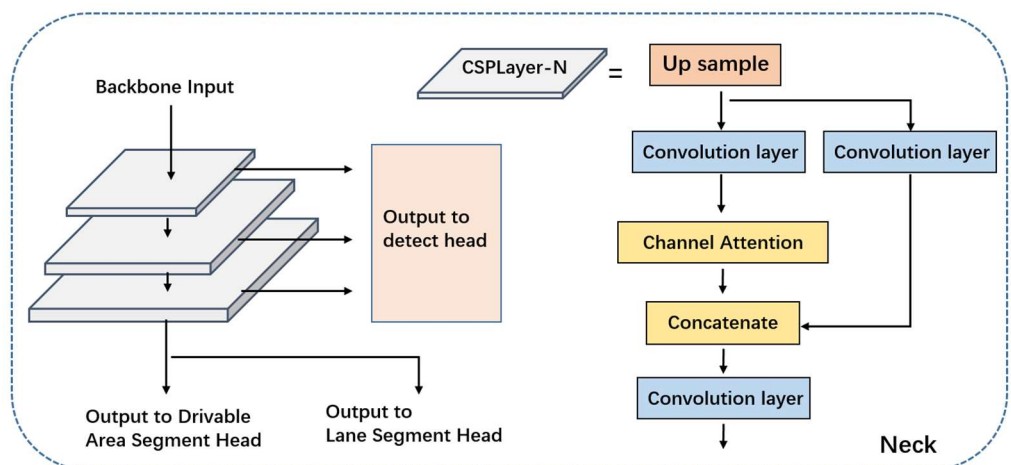

**Figure 4.** Neck network architecture.

For each grid in the multiscale feature maps of the object-box regression branch, anchors were assigned to three different aspect ratios, and the sensing head predicted position offset, height, and width of the scale and predicted the likelihood of each class and the corresponding confidence value, referring to the practice of NAS-FPN in the first two layers of the detection task head, using the SepBN head, sharing the values of the convolution weights across different layers but computing the batch normalization separately. The structure of the sensing task header is composed of three sensing modules in parallel, as shown in Figure 5. Note that the two outputs of the detection module correspond to the box loss and the classification loss, respectively.

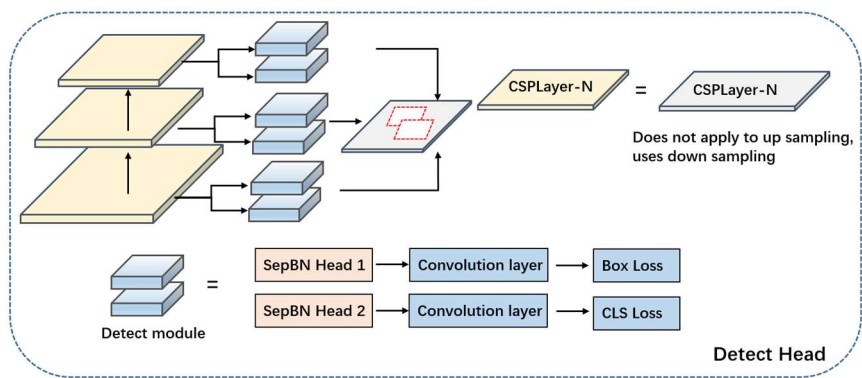

**Figure 5.** Detect head network architecture.

In our approach, both the drivable zone and lane line segmentation are performed in task heads with similar, but separate, structures. Unlike YOLOPV2, the features for both tasks come from the last layer of the neck. We found that segmenting the drivable region was difficult in the ApolloScape dataset [30] and that features extracted from the deeper

layers of the network were necessary to segment the drivable region. Shallow features do not improve prediction performance. In this way, the segmentation head of the drivable region is connected to the end of the FPN for the input of the deeper feature maps, similar to that in YOLOP. In the case of lane line segmentation, the task branches also branch to the end of the FPN in order to extract deeper features, as the lane lines are often very fine and difficult to detect in the input image. Therefore, as shown in Figure 6, the main module in the windowing network is CSPLayer-US, in which the top sampling operation is mainly performed in the feature map. The decoder stage of lane line detection also makes use of deconvolution to further improve performance.

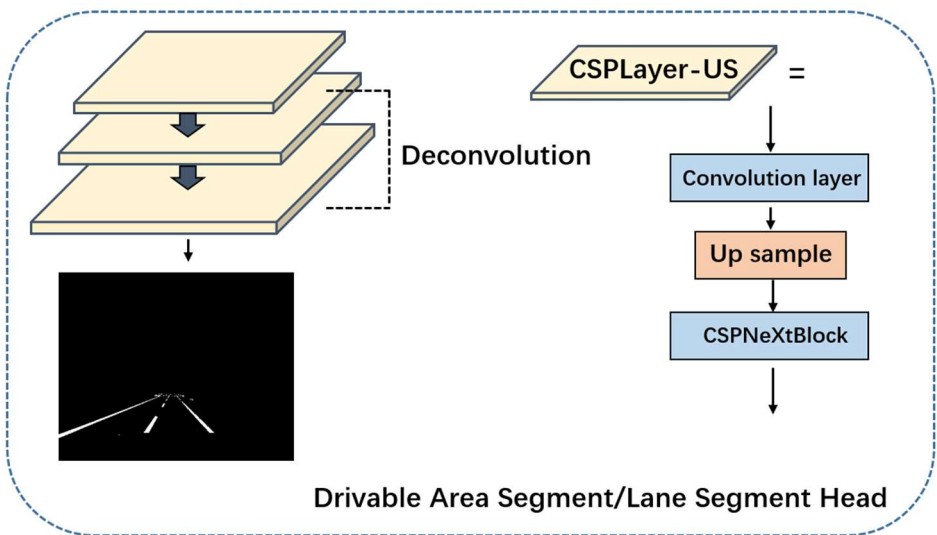

**Figure 6.** Segmentation head network architecture.

### 3.4. Loss Function Design

Given that we have three task decoders in our network, our multi-task loss contains three components. As can be seen in Equation (1), the target detection loss is a weighted sum of the detection frame category loss, the detection frame object loss, and the detection frame size loss.

$$\mathrm{L}_{det} = \alpha_1 L_{\mathrm{class}} + \alpha_2 \mathrm{L}_{obj} + \alpha_3 \mathrm{L}_{box} \tag{1}$$

where $L_{class}$ and $\mathrm{L}_{obj}$ are part of the focal loss, which is proposed to solve the problem of unbalanced distribution of a large number of positive and negative samples in the candidate frame of the target detection. Focal loss takes the perspective of sample difficulty classification so that the loss focuses on discriminating difficult samples. $L_{class}$ is used to distinguish which loss of the detection frame belongs to the category to be classified, and $\mathrm{L}_{obj}$ indicates for the loss of the detection frame belonging to the background or the object. $\mathrm{L}_{box}$ is derived from $\mathrm{L}_{CIoU}$; it takes into account the center distance, overlap, scale, and aspect ratio of the predicted frame to the actual annotated frame.

We have introduced the Softmax method of label processing of $L_{cls}$, in Equation (2). The existing methods usually use binary direct labels to calculate the classification loss, which leads to a high classification score for the boundary box prediction, so the prediction of its detection frame boundary is unreasonable.. However, the prediction with a low classification score but high detection frame score obtained a lower-than-expected loss score. The soft label method is helpful to solve this problem.

$$L_{cls} = CE(P, Y_{softmax}) \times (Y_{softmax} - P)^2 \tag{2}$$

Inspired by GFL [31], this design uses the predicted value of IoU between the prediction frame and the true frame as a soft-label $Y_{softmax}$ to train the classification loss in the target detection head. The soft classification cost in the assignment not only re-weighs

the matching loss of different regression quality prediction frames but also avoids the unbalanced and confusing feedback caused by direct binary labelling.

Division of the driving area $L_{ll-seg}$ and lane line division $L_{ll-seg}$ loss includes logarithmic cross entropy loss $L_{ce}$, the purpose of which is to minimize the classification error between the network output pixel and the target. It is worth mentioning that IoU loss in Equation (3) shows the following:

$$L_{IoU} = 1 - \frac{TP}{TP + FP + FN} \tag{3}$$

Because it is particularly effective for the prediction of sparse categories of lane lines, it is added to $L_{ll-seg}$. $L_{da}$ and $L_{ll-seg}$ defines Equations (4) and (5), respectively.

$$L_{da-seg} = L_{ce} \tag{4}$$

$$L_{ll-seg} = L_{ce} + L_{IoU} \tag{5}$$

To sum up, our final loss is the weighted sum of the three parts, as shown in Equation (6).

$$L_{all} = \gamma_1 L_{det} + \gamma_2 L_{da-seg} + \gamma_3 L_{ll-seg} \tag{6}$$

Among them, $\alpha_1, \alpha_2, \alpha_3, \gamma_1, \gamma_2, \gamma_3$ are adjustable weighting factors for each component loss value. By default, these parameters are set to $\alpha_1 = 1, \alpha_1 = 3, \alpha_1 = 1, \gamma_1 = 1, \gamma_2 = 2, \gamma_3 = 2$.

### 3.5. Algorithm Details Implementation

We tried different methods to train the model, such as the step-by-step method, task type step-by-step method, end-to-end method, and pre-training paradigm. In these experiments, we found that using the ImageNet dataset [32] as a pre trained backbone model followed by downstream task fine-tuning can achieve high accuracy. The process of our end-to-end direct training method is shown in Algorithm 1.

---

**Algorithm 1.** The end-to-end direct training method

---

**Input:** Complete data set : $D_m$; single batch size : $K_m$;
the target neural network $\mathcal{F}$; random initialization parameters $\theta_0$;
maximum number of iterations: $T$; learning rate : $Lr$.
**Output:** Well $-$ trained network : $\mathcal{F}(\mathbf{x}; \theta_0)$
**1 For** t = 1 in $T$ **do**
// There are three tasks
**2**      **For** $m$ = 1 in 3 **do**
**3**         Randomly divide dataset $D_m$ into sets with $c = D_m / K_m$
**4**         $\mathcal{B}_m = \{\mathcal{J}_{m,1}, \mathcal{J}_{m,2}, \dots, \mathcal{J}_{m,c}\}$**;**
**5**      **End**
**6**      Merge all small batch samples $\overline{\mathcal{B}} = \mathcal{B}_1 \cup \mathcal{B}_2 \cup \mathcal{B}_3$;
**7**      Random sorting $\overline{\mathcal{B}}$;
**8**      **Foreach** $\mathcal{J}$ **in** $\overline{\mathcal{B}}$ **do**
**9**         Calculate the $\mathcal{J}$ of samples loss $L(\theta_n)$;
**10**        Merge the losses on each task $L_{all}(\mathcal{F}(\mathbf{x}; \theta_n))$; // x is the input image
**11**        Update parameters : $\theta_t \leftarrow \theta_{t-1} - Lr \cdot \nabla_\theta L_{all}(\theta_n)$
**12**      **End**
**13 End**
**14 Return** Trained networks $\mathcal{F}(\mathbf{x}; \theta)$

---

## 4. Experimental Evaluation

This section describes the data set setting and parameter configuration of our experiment. The model training in this paper uses two RTX3080 GPUs and a torch 1.10 environment. All reasoning experiments are based on the configuration environment of RTX3060 GPU and torch 1.10.

### 4.1. Data Sets

For the experimental research, we used ApolloScape as our benchmark dataset, which is a publicly available dataset consisting of challenging driving scenarios. The dataset contains 1 million frames from various camera viewpoints on top of the car, 800 thousand frames of LiDAR point cloud data, and 1000 km of city traffic collection trajectories. The dataset has been referenced for a variety of autonomous driving tasks, such as binocular and monocular depth map generation challenges, traffic participant trajectory prediction, and more. The ApolloScape dataset also supports eight vision tasks.

In comparison to other popular driving datasets, such as Cityscapes, Camvid, and BDD100, ApolloScape provides a larger number of data types, such as laser radar and stereo cameras. This facilitates further new developments for downstream tasks, such as a predictive algorithm for the generation of bird's-eye-views (BEV) [33] from a higher perspective of autonomous driving. As in other research, we extracted a target detection dataset from the scene segmentation dataset and combined it with the original scene segmentation dataset as well as the lane detection dataset to form the dataset that is used in this paper. The dataset is split into a training set of 70,000 images, a validation set of 10,000 images, and a test set of 20,000 images. Figure 7 shows a portion of the dataset, which is, from left to right, the original camera input image, the scene segmentation annotation, the lane annotation, and the extracted target detection annotation.

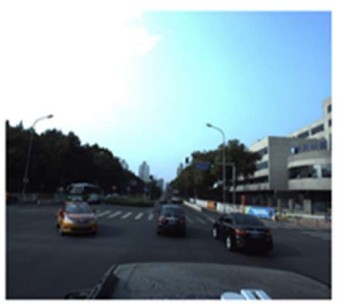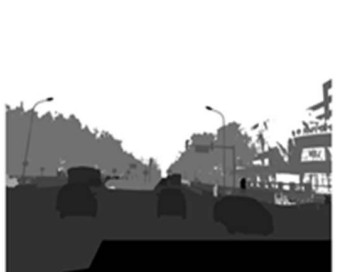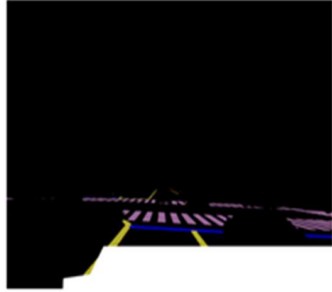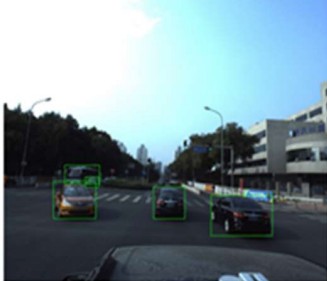

**Figure 7.** Visualization of dataset annotations.

### 4.2. Experimental Process

Our experimental procedure only had a multiple iteration approach, following a global update of hyperparameters and partial structures after one complete training of the model, gradually completing the full method improvement. The experimental flowchart is shown in Figure 8.

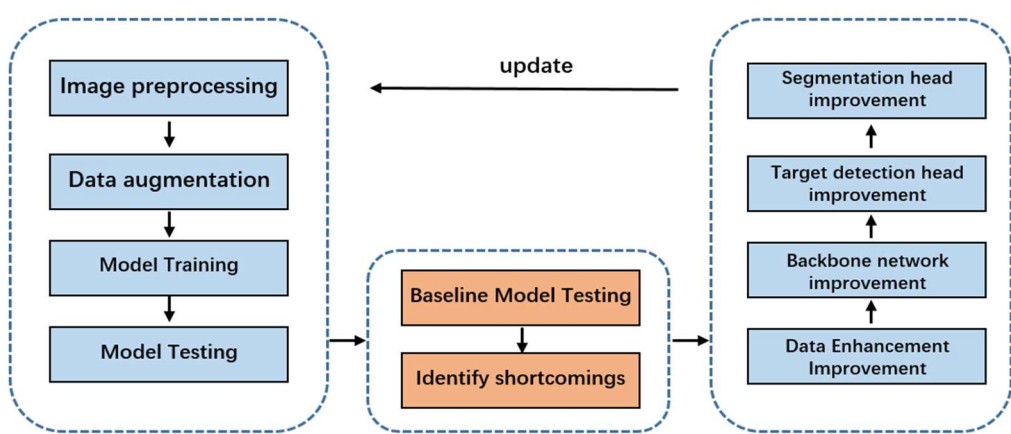

**Figure 8.** Experimental flowchart.

### 4.3. Training Methods

During the training process, we used the cosine annealing strategy to tune the learning rate, where the optimizer and initial learning rates β1 and β2 were fixed at 0.001, 0.879, and 0.999, respectively. In order to achieve faster and better network convergence, the learning rate was tuned by preheating and cosine annealing. A total of 300 training iterations were performed. During the training and testing phase, the image in the ApolloScape dataset was a crop from $3384 \times 2710 \times 3$ to $2710 \times 2710 \times 3$, then scaled down to $640 \times 640 \times 3$ in a proportional manner.

The loss curve obtained from the training is shown in Figure 9. The abscissa represents the number of training steps, and the vertical coordinate is the composite loss value during training. As shown in Figure 9, the loss value decreases rapidly at the beginning of the training process, indicating that the selection of the learning rate is appropriate. When the iteration reaches 1250 steps, the curve tends to be flat, indicating that the model has converged. Our model training loss is lower than YOLOP, and the validation effect is also better.

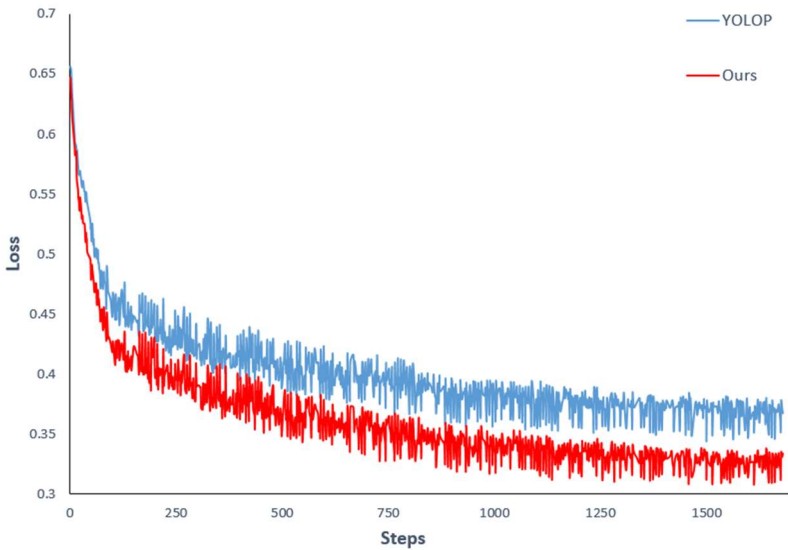

**Figure 9.** Loss curve during training.

It should be noted that we used the number of steps instead of epochs to draw the line graph of the loss value, because we found that it will change greatly in epochs in the experiment. In order to facilitate observation, we used the number of steps to save the results.

### 4.4. Comparison of Experimental Results

Table 2 shows how our model compares with YOLOP and HybridNets. All tests were performed in the same experimental setting and with the same evaluation metrics.

**Table 2.** Comparison of network parameters and reasoning speed.

| Model | Input Shape | Parameters | Reasoning Time (ms) |
|---|---|---|---|
| YOLOP | $640 \times 640$ | 7.92 M | 11.1 |
| HybridNets | $640 \times 640$ | 12.84 M | 21.7 |
| Our method | $640 \times 640$ | 6.3 M | 7.6 |

It can be seen in Table 2 that our method has a parameter size of 6.3 million (represented by 6.3 M), which is smaller than the 2 baseline models and requires less processor computation. The computing performance of our method is better because of the efficient network design and sensible use of the GPU memory strategy.

Unlike the baseline algorithm YOLOP, mAP50:95 and recall are used here as evaluation metrics; mAP50:95 refers to the value of IOUs taken from 50% to 95% in steps of 5%, and then the mean of the average precision among these IOUs is calculated. Our network achieved a higher mAP50:95 and recall rate, as shown in Table 3.

**Table 3.** Traffic Target Detection Results.

| Model | mAP50:95 | Recall |
|---|---|---|
| MultiNet | 59.1 | 83.2 |
| DLT-Net | 68.3 | 86.4 |
| HybridNets | 75.4 | 92.3 |
| YOLOV5s | 77.9 | 94.3 |
| YOLOP | 76.8 | 93.8 |
| Our method | 78.6 | 94.6 |

In the drivable region segmentation task, we used mIoU to evaluate the segmentation performance of different models. As can be seen from the table, our network had better performance, as shown in Table 4 below.

**Table 4.** Results of driving area segmentation.

| Model | Drivable mIoU |
|---|---|
| DLT-Net | 79.2 |
| HybridNets | 95.4 |
| YOLOP | 96.0 |
| Our method | 97.1 |

In the ApolloScape dataset, the tracks consist of multiple labels and, thus, must be preprocessed. By using lane lines and zebra crossings as lane line labels, we also use the pixel accuracy and the lane throughput as evaluation metrics. Pixel accuracy represents the number of correctly classified pixels in the lane line segmentation task divided by the number of all pixels. Table 5 shows that our network achieved the highest value in terms of precision.

**Table 5.** Lane detection results.

| Model | Lane mIoU | Accuracy (CCP/AP) |
|---|---|---|
| MultiNet | 55.9 | 68.8 |
| DLT-Net | 69.4 | 70.9 |
| HybridNets | 85.3 | 77.6 |
| YOLOP | 85.3 | 76.5 |
| Our method | 86.8 | 78.7 |

*4.5. Visualization*

Figures 10–13 display the visualized comparison results of YOLOP and our model on the ApolloScape dataset under different road tests. Figure 10 shows the test results on Road 1. The first column is the ground truth labeled image, and the second column lists YOLOP's effects. There are several predicted wrong drivable areas in the first scene. In Figure 11, YOLOP's prediction result has missing detection boxes of small objects and the wrong segmentation of the drivable area. In the test results on Road 3, in Figure 12, missed detection of lanes are found. In the test results of Road 4, in Figure 13, different degrees of false alarms are marked at the left turn, of which YOLOP has false detection at the far intersection. Based on those results, the right column shows our results, which demonstrates the better performance of our network in various scenarios.

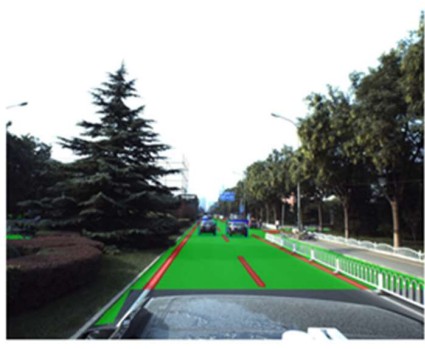 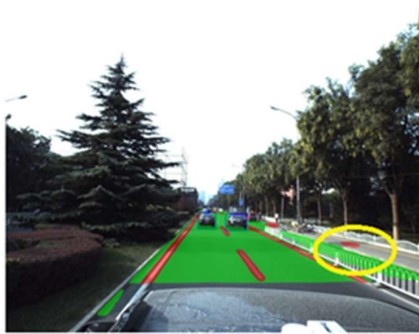 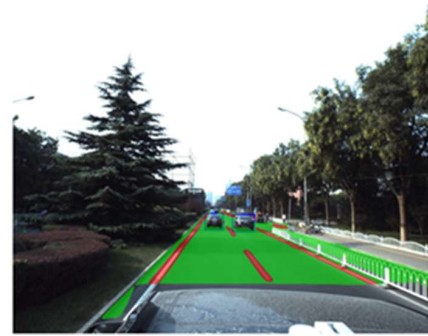

**Figure 10.** Road 1 scenario test results.

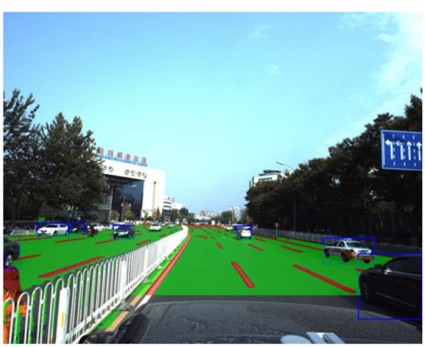 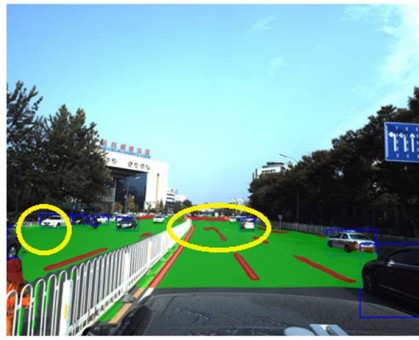 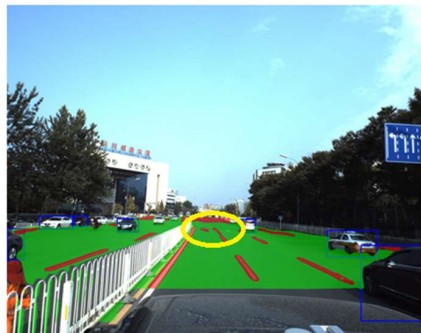

**Figure 11.** Road 2 scenario test results.

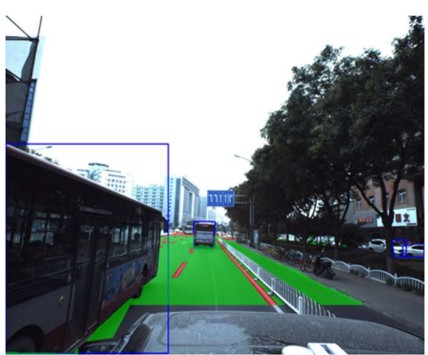 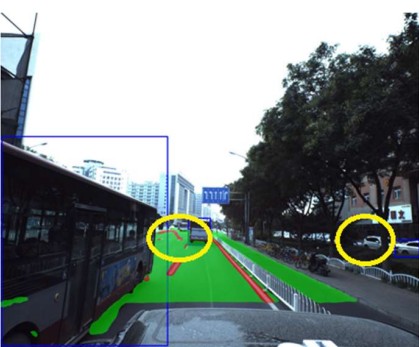 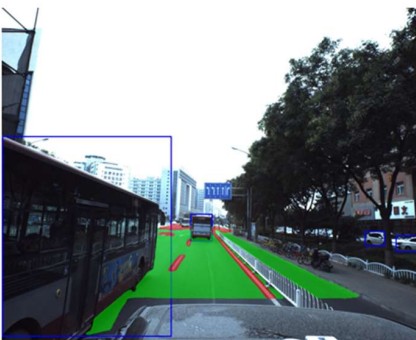

**Figure 12.** Road 3 scenario test results.

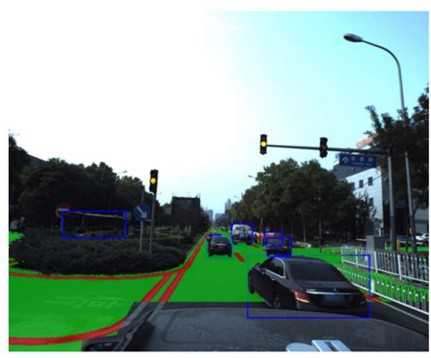 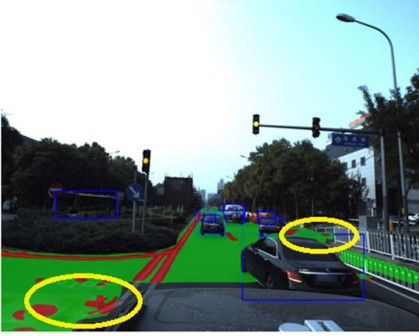 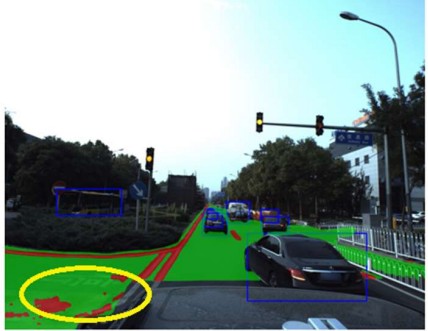

**Figure 13.** Road 4 scenario test results.

In Figures 10–13, the blue boxes represent the results of traffic target detection, the yellow circles represent the differences between the test results of different models, the

green areas represent the segmentation results of the driveable area, and the red lines represent the results of lane line detection.

*4.6. Ablation Studies*

We conducted various modifications and improvements to the baseline network and corresponding experiments. Table 6 shows some of the changes we made and their corresponding improvements in experiments. The experimental results show that each of our improvements bring a corresponding performance improvement. The main reasoning speed improvement comes from the improvement of the backbone network, and the improvement of target detection accuracy is mainly from the introduction of SepBN Head and the optimization of the loss function. In addition, the accuracy of lane line detection is also crucial to the improvement of deconvolution.

**Table 6.** Evaluation of efficient experiments.

| Method | Reasoning Speed (FPS) | Object Detection mAP50:95 | Object Detection Recall | Drivable Area mIoU | Lane Detection Accuracy |
|---|---|---|---|---|---|
| YOLOP (Baseline) | 90 | 76.8 | 93.8 | 96.0 | 76.5 |
| +CSPNeXt | 126 | 77.5 | 94.3 | 96.1 | 77.3 |
| +MosaicandMixup | 124 | 778 | 94.8 | 96.1 | 77.5 |
| +Convtranspose2d | 136 | 76.8 | 94.8 | 96.1 | 78.7 |
| +SepBN Head | 131 | 78.6 | 94.6 | 96.1 | **78.7** |

## 5. Conclusions

In this work, we propose an optimal network by using CSPNeXts to implement a backbone network and introduce soft labels and shared weights in target detection. We also used advanced data augmentation techniques for this real-time multitask learning network to improve the performance of the algorithm in autonomous driving scenarios. We conducted experiments on the challenging dataset of ApolloScape. Our network achieved the best performance in all three tasks: mAP50:95 of 0.786 for the target detection task, mIoU of 96.1 for the drivable area segmentation task, and accuracy of 78.7 for lane detection. It improved both accuracy and speed compared to the baseline model. In model inference, the frame FPS execution speed improved to 131 FPS on the NVIDIA RTX 3060 device, higher than the 90 FPS of YOLOP under the same experimental conditions. Overall, the network parameters were reduced by 24%, reasoning speed was increased by 45%, and the required computational power was also lowered.

This work can be used in edge devices with limited resources to perform multitasks, which can be applied in different scenarios, such as rural roads, unstructured roads, and special roads. Moreover, it also can be used in computer vision for autonomous driving to improve the safety of driving by reducing accidents caused by human error.

In the future, we will strive to apply autonomous driving multitask learning to more diverse datasets and different weather scenarios and add new tasks such as depth estimation, pedestrian posture prediction, and traffic object prediction.

**Author Contributions:** Conceptualization, X.Z. and G.Y.; methodology, X.Z. and G.Y.; software, X.Z. and C.L.; validation, X.Z., C.L. and P.Z.; formal analysis, X.Z. and C.L.; investigation, C.L.; resources, G.Y. and X.Z.; data curation, X.Z. and C.L.; writing—original draft preparation, X.Z., C.L., G.Y. and P.Z.; writing—review and editing, X.Z., C.L., G.Y. and P.Z.; visualization, C.L.; supervision, X.Z. and C.L.; project administration, X.Z.; funding acquisition, X.Z. and G.Y. All authors have read and agreed to the published version of the manuscript.

**Funding:** This research was funded by Young and Middle-aged teachers in Fujian Province under grant number JAT210674 and by the Natural Science Foundation of Fujian Province under grant number 2021J01865, 2021J01866.

**Data Availability Statement:** We include a data availability statement with all Research Articles published in an MDPI journal.

**Conflicts of Interest:** The authors declare no conflict of interest.

## Nomenclature

| Term | Term Definition |
| --- | --- |
| ADAS | Advanced driver assistance systems, a technology that assists drivers in the driving process, found in reference [1]. |
| ApolloScape | An open-source dataset for autonomous driving research and computer vision tasks, found in reference [30]. |
| BasicBlock | A building block of convolutional neural networks used for feature extraction. |
| BEV | Bird's eye view, a perspective view used in computer vision to represent a top-down view of an object or environment, found in reference [33]. |
| BiFPN | Bidirectional feature pyramid network, a network used for object detection tasks, found in reference [22]. |
| CCP/AP | Pixel accuracy represents the number of correctly classified pixels in the lane line segmentation task divided by the number of all pixels |
| CenterNet | An object detection framework that uses keypoint estimation to detect objects, found in reference [3]. |
| ConvNeXt | A type of convolutional neural network architecture that improves upon traditional convolutional layers by using grouped convolutions, found in reference [27]. |
| CSPdarknet | A lightweight deep learning framework for computer vision tasks. |
| CSPNeXt | A deep neural network model for image classification and object detection tasks. |
| CSPNeXtBlock | A building block used in CSPNeXt architectures. |
| CSPLayer-T | A layer used in CSPNeXt architectures. |
| FCN | Fully convolutional network, a type of neural network commonly used for semantic segmentation tasks, found in reference [15]. |
| FPN | Feature pyramid network, a neural network used for object detection and semantic segmentation tasks. |
| GFL | Generalized focal loss, a loss function used in object detection tasks, found in reference [31]. |
| HybridNet | A neural network architecture that combines both convolutional and recurrent layers, found in reference [20]. |
| LaneNet | A neural network used for lane detection and segmentation tasks, found in reference [19]. |
| MultiNet | A multitask learning framework used for various computer vision tasks, found in reference [13]. |
| PAN | Path aggregation network, a type of neural network used for semantic segmentation tasks. |
| PolyLaneNet | A neural network used for lane marking detection in autonomous driving. |
| PSPNet | Pyramid scene parsing network, a neural network used for semantic segmentation tasks, found in reference [11] |
| RepLKNet | A neural network architecture for object detection and instance segmentation tasks. |
| RetinaNet | An object detection framework that uses focal loss to address class imbalance issues, found in reference [2] |
| R-CNN | Region-based convolutional neural network, an object detection framework, found in reference [4] |
| RTMDet | Real-time multi-person detection, an object detection framework used for real-time multi-person detection, found in reference [27] |
| SAD | Learning lightweight lane detection convolutional models through self-attentive refinement, found in reference [16] |
| SegNet | A neural network used for semantic segmentation tasks, found in reference [10] |
| SepBN Head | Separable batch normalization head, a type of normalization technique used in convolutional neural networks. |
| Softmax | A function used to convert a vector of numbers into probabilities that sum to one. |
| SPP | Spatial pyramid pooling, a method used to handle variable-sized inputs in neural networks, found in reference [29] |
| Swin-T | Swin Transformer, a transformer-based neural network architecture commonly used for computer vision tasks, found in reference [25] |
| UNet | A neural network architecture used for image segmentation tasks, found in reference [9] |
| Vit | Vision transformer, a transformer-based neural network architecture commonly used for computer vision tasks, found in reference [24] |
| YOLO | You only look once, an object detection framework that predicts bounding boxes and class probabilities directly from the input image, found in reference [6] |
| YOLOP | You only look once for panoptic driving perception, a multitask learning model for autonomous driving, found in reference [19] |

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
