# Peer review of "Visual Multitask Real-Time Model in an Automatic Driving Scene"

_electronics, doi:10.3390/electronics12092097_

Round 1

Reviewer 1 Report

1. Authors have employed two distinct methods for defining abbreviations, such as Self-Attention Distillation (SAD) versus spatial pyramid pool module (SPP), and consistency must be maintained.

2. Despite the authors' efforts to showcase the manuscript's novelty, I urge them to devote more attention to this aspect to capture readers' interest.

3. Please provide a detailed experimental flowchart to enable other researchers to easily reference the proposed method.

4. Could you explain the rationale behind selecting the parameters and their ranges for the network architecture? What numerical parameters enabled the architecture to yield consistent solutions?

5. The paper lacks a Nomenclature section. It would be beneficial if the authors could include one, separating Roman and Greek letters and listing them alphabetically.

6. Since the paper lacks comprehensive descriptions of the proposed algorithm, it would be beneficial to include some pseudocode or algorithm steps.

7. What does "M" mean in Table 2?

8. Please include units/metrics, such as "accuracy," in Table 5.

9. Are reasoning speed and inference the same? If so, it is preferable to maintain consistency.

10. The conclusions are insufficient. It would be beneficial to highlight the broader applications of the techniques, emphasizing their global perspective in addressing the problem. I recommend revising it in line with the paper's most compelling conclusion.

11. Please add more precise research directions that can be pursued based on your paper.

Reviewer 2 Report

This paper proposes a network with better overall performance, targeting the three tasks of traffic target detection, drivable road segmentation, and lane detection that need to be performed simultaneously by visual perception in automatic driving scenes and their real-time requirements. Experimental results were performed on the ApolloScape dataset, showing that the proposed network outperforms in terms of generalization with a more effective network structure. In addition, advanced network enhancement techniques, such as mosaic padding & image mixing processing during data pre-processing, and optimizing the loss function of the target detection task head, are used to improve the accuracy and speed of the model.

Some minor comments:

- p. 4, at section Backbone, could the authors better explain the structure diagram of Darknet Block (Figure 2)?

-p. 9, the authors state "Thereby facilitating further downstream development such as the search for BEV generation algorithms", this passage is not clear, could the Authors better explain? 

- p. 10, the authors write "We see that our network has a faster performance. Because of efficient network design and a sophisticated memory allocation strategy ", I think there is a punctuation error
